# Genome-wide association study reveals *GmFulb* as candidate gene for maturity time and reproductive length in soybeans (Glycine max)

Diana M. Escamilla[1], Nicholas Dietz[2], Kristin Bilyeu[3], Karen Hudson[4], Katy Martin Rainey[1]*

1 Department of Agronomy, Purdue University, West Lafayette, Indiana, United States of America, 2 Division of Plant Science and Technology, University of Missouri, Columbia, Missouri, United States of America, 3 Plant Genetics Research Unit, United States Department of Agriculture (USDA)−Agricultural Research Service (ARS), Columbia, Missouri, United States of America, 4 USDA-ARS Crop Production and Pest Control Research Unit, West Lafayette, Indiana, United States of America

* krainey@purdue.edu

**Data Availability Statement:** Derived data supporting the findings of this study are available in the Github repository (https://github.com/DianaME/Dataset-Soybean-Maturity).

## Abstract

The ability of soybean [*Glycine max* (L.) Merr.] to adapt to different latitudes is attributed to genetic variation in major *E* genes and quantitative trait loci (QTLs) determining flowering time (R1), maturity (R8), and reproductive length (RL). Fully revealing the genetic basis of R1, R8, and RL in soybeans is necessary to enhance genetic gains in soybean yield improvement. Here, we performed a genome-wide association analysis (GWA) with 31,689 single nucleotide polymorphisms (SNPs) to detect novel loci for R1, R8, and RL using a soybean panel of 329 accessions with the same genotype for three major *E* genes (*e1-as/E2/E3*). The studied accessions were grown in nine environments and observed for R1, R8 and RL in all environments. This study identified two stable peaks on Chr 4, simultaneously controlling R8 and RL. In addition, we identified a third peak on Chr 10 controlling R1. Association peaks overlap with previously reported QTLs for R1, R8, and RL. Considering the alternative alleles, significant SNPs caused RL to be two days shorter, R1 two days later and R8 two days earlier, respectively. We identified association peaks acting independently over R1 and R8, suggesting that trait-specific minor effect loci are also involved in controlling R1 and R8. From the 111 genes highly associated with the three peaks detected in this study, we selected six candidate genes as the most likely cause of R1, R8, and RL variation. High correspondence was observed between a modifying variant SNP at position 04:39294836 in *GmFulb* and an association peak on Chr 4. Further studies using map-based cloning and fine mapping are necessary to elucidate the role of the candidates we identified for soybean maturity and adaptation to different latitudes and to be effectively used in the marker-assisted breeding of cultivars with optimal yield-related traits.

**Funding:** Funding for this study was obtained by KR from the Indiana Soybean Alliance (https://indianasoybean.com/) from several grants. The funders had no role in study design, data collection and analysis, the decision to publish, or the preparation of the manuscript.

**Competing interests:** The authors have declared that no competing interests exist.

## Introduction

Soybean's (*Glycine max*) high protein, oil, and carbohydrate content make it a valuable crop worldwide, with various applications in many industries [1–3]. Soybean plants are photoperiod-sensitive and flower under short-day conditions. Soybeans grow across 50°N to 35°S latitudes, and the critical day length period for flowering decreases progressively from higher to lower latitudes [4–8]. Soybean varieties are classified into 13 maturity groups (MG000 to MGX) based on their region of adaptation and day length requirements [8, 9]. When grown outside the optimal latitudinal range, soybeans will flower and mature late in higher latitudes or too early in lower latitudes, reducing biomass and yields [8, 9]. Within each latitudinal range, or region, there is also variation in the length of the growing season, and soybeans are classified as early, mid-, or full season [9].

The change from vegetative to reproductive growth and then to seed maturation are critical developmental switches in soybean influenced and determined by genotype, photoperiod, temperature, elevation, and management [10]. Identifying and understanding the molecular mechanisms and genetic architecture underlying flowering and maturity time is crucial for improving soybean adaptation and yield across differing and variable environments. A number of '*E* genes' and quantitative trait loci (QTLs) contribute to flowering and maturity time through a photoperiod mediated response with different allelic combinations of *E* genes determining soybean adaptation to specific latitudes [8, 11–13]. Eleven maturity loci, *E1* to *E11*, have been reported to control flowering and maturity. Soybean photoperiod sensitivity decreases with the number of recessive alleles [8], while dominant alleles confer late flowering and late maturity except for *E6/J* and *E9* genes [13, 14]. Soybean stem growth also plays a crucial role in flowering time and maturity. The two known genes regulating stem growth are *Dt1*, with the *Dt1Dt1* genotype producing indeterminate stem growth, and *Dt2*, with the *Dt2Dt2* genotype producing semi-determinate plants in the presence of the genotype *Dt1Dt1*. In contrast, the *dt2dt2* genotype produces indeterminate stem growth in the presence of *Dt1Dt1*; however, if the *dt1dt1* genotype is present, the phenotype is determinate [12, 15, 16].

*E1* and *E2* genes delay flowering by suppressing the expression of *GmFT2a* and *GmFT5a*, which are flowering inducers [10, 17–21]. In addition to the *E1* locus, there are two homologous genes, *E1la* and *E1lb*, which repress the expression of *GmFT2a* and *GmFT5a* independently of *E1* [21, 22]. Known variant missense and nonfunctional alleles for the *E1* gene include the *e1-as*, *e1-b3a*, *e1-re*, *e1-fs*, *e1-p*, and *e1-nl* alleles [23–25]. *E3* and *E4* are phytochrome A genes sensitive to red-to-far-red light ratios that promote late flowering under long-day conditions by regulating the expression of *E1* and suppressing *GmFT2a* and *GmFT5a* [22, 23, 26–29]. The *E3* locus has two functional (*E3-Ha* and *E3-Mi*) and three nonfunctional (*e3-tr*, *e3-ns*, and *e3-fs*) alleles described [26, 29] and the *E4* locus has one functional and five nonfunctional alleles (*e4*, *e4-oto*, *e4-tsu*, *e4-kam*, and *e4-kes*) described [27, 28]. The *E5* locus has not been mapped [30, 31]. *E6* and *J* are alleles of the same locus; where *E6* promotes early flowering, and *J* confers a long-juvenile phenotype by delaying flowering under short-day conditions [7, 32–34]. There is not yet a clear understanding of the molecular mechanism for flowering regulation at the *E7* and *E8* is *E1Lb* loci [21, 35–38]. *E9* and *E10* are flowering loci *GmFT2a* and *GmFT4*, respectively, with dominant genotypes promoting earlier flowering than the recessive genotypes [13, 39]. The most recently-reported maturity gene is *E11* which induces earlier flowering when a single dominant allele is present [40].

Days to flowering (R1) and maturity (R8), and duration of flowering to maturity (RL) are critical traits determining soybean adaptation, seed quality, and yield [11, 41, 42]. Most studies focus on understanding the impact of *E* genes on R1 and R8, while there are few studies on the impact of *E* genes on post-flowering and RL [11, 29, 43]. Maturity genes *E1* to *E4* have the

most significant effect on R1, R8, and photoperiod sensitivity, controlling as much as 62–66% of the variation in R1 [24, 44, 45]. *E1* has a major effect on R1 and is a crucial regulator of flowering in soybean [46]. Studies on the allelic variation of *E1* to *E4* in different soybean populations reveal a complex genetic control underlying R1 and suggest the existence of unknown genes with roles in R1 [8, 24, 29, 44, 47, 48]. Many other QTLs that have been discovered show evidence of influencing R1 and R8; however, based on their map position, some correspond to the known *E* genes [41, 43, 49].

An effective way to detect associations between single nucleotide polymorphisms (SNPs) and the trait of interest is through genome-wide association studies (GWAS). GWAS commonly estimate the marginal effect of individual SNPs by single-locus analysis, which only detects the SNPs that have relatively large effects, leaving out the SNPs with minor effects [50–54]. Fully uncovering the genetic architecture of important traits is a complicated task that requires careful consideration of the study design, testing populations, data collection, and analysis. At the same time, there is evidence that major-effect loci are less influenced by genetic background than minor-effect loci [55], and it is probable that major *E* genes with large effects on R1 and R8 alter, mask or hide the effect of unknown minor *E* genes. In order to understand the genetic regulation of R1 and R8 beyond what is already known for the eleven major genes, researchers studied causal genetic interactions through epistasis and populations sharing the same genotype for major *E* genes [54, 56, 57]; however, the molecular function of R1- and R8-related genes is still elusive with many unknown loci yet to discover.

Although there is much progress in identifying new maturity loci and understanding the genetic architecture of R1, R8, and RL, the known loci explain only a portion of variation, leading many to question how to explain the remaining variation. Identifying novel loci for R1, R8 and RL may improve our understanding of soybean adaptation to different latitudes, which will facilitate maximizing yield for specific environments. Here, we proposed to investigate the genetic architecture of R1, R8, and RL beyond major E genes (*E1* to *E3*) by high-throughput genetic markers and association analysis on a soybean panel from MG III and IV sharing the same allelic combination '*e1-as/E2/E3*' for major genes.

## Materials and methods

### Plant material and experimental design

To assemble the diversity panel, we selected 329 soybean accessions predicted to share the same allelic combination for major *E* genes, *e1-as/E2/E3*, from the soybean germplasm collection using the GRIN data explorer (https://www.soybase.org/grindata/). This tool facilitates searches of the GRIN descriptor data [58]. The *e1-as* allele is a missense, recessive allele for the *E1* gene [23], while *E2* and *E3* are functional alleles. The selected 329 accessions were from MG III and IV and include 220 improved cultivars and 109 breeding lines with either indeterminate (*Dt1Dt1/dt2dt2*) or semi-determinate (*Dt1Dt1/Dt2Dt2*) stem growth and predicted to be resistant to shattering (Fig 1). Descriptor data available in the GRIN data explorer for shattering and *E1*, *E2*, and *E3* genes, were imputed. The alleles at the shattering (*Pdh1*) locus were imputed in a previous study performing a genome-wide association study (GWAS) using the *Pdh1* allele status as a phenotype [59]. They identified a highly associated marker in the SoySNP50K array that they later used to predict the *Pdh1* allele status in the GRIN collection. To predict the *E1*, *E2*, and *E3* genes, researchers studied a genomic dataset of 406 soybean genotypes, including cultivars, landraces, and wild species (*Glycine soja*) [58]. They utilized GWAS and direct genotype information to select associated markers that could accurately predict the allele status of *E1*, *E2*, and *E3* in the GRIN collection.

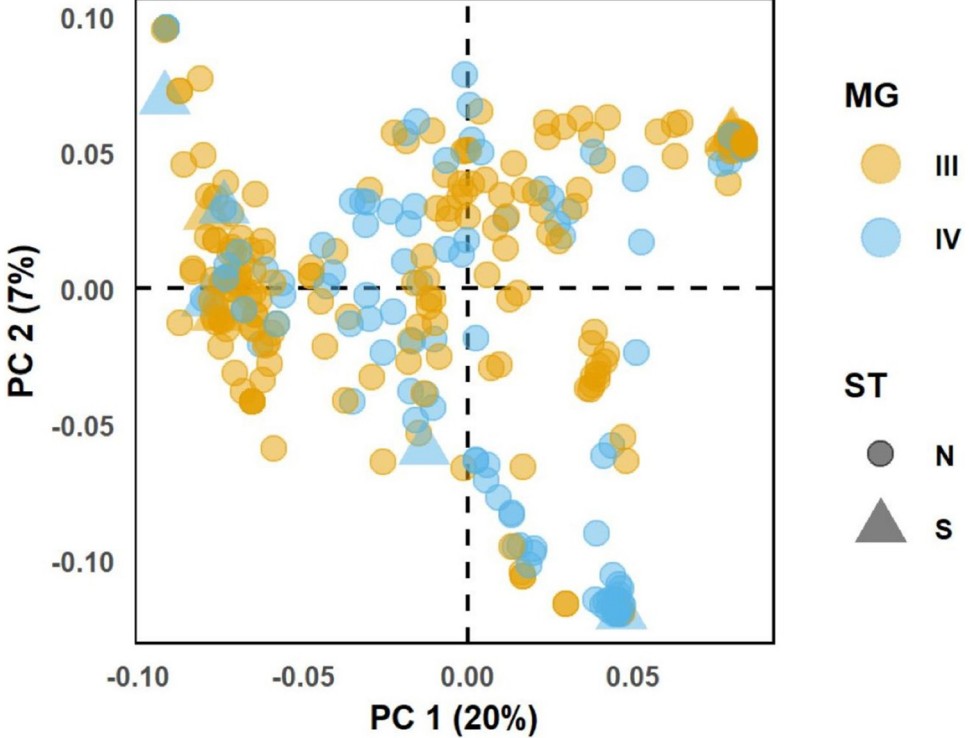

**Fig 1. Principal component analysis (PCA) biplot of three hundred twenty-nine *G*. max USDA accessions.** Individuals are represented with different colors and shapes according to maturity group (MG) and stem termination type (ST). N-indeterminate, S- semideterminate.

We grew all accessions in single row plots in Indiana (ACRE-IN) and Missouri (Columbia-MO) in 2017 and 2018. In 2017, single row plots were 0.3 m long with 0.3 m spacing among plots and 10 seeds per plot. The single row plots in 2018 were 1.83 m in length with 0.72 m spacing among plots and 50 seeds per plot. To control for potential heterogeneity of experimental units, we evenly distributed maturity checks, which included 194Dremut #3021, DSN11-03148 (MGIV), DSN11-06152 (MGIII), DSN11-12073 (MGII), and IA3023 (MGIII), over the field experiment area for each location to facilitate spatial correction of field plot variation. In 2019 and 2020, we grew the accessions in four-row plots, two locations (ACRE Farm, West Lafayette, IN, and Ag Alumni Seeds, Romney, IN), and two replications in a completely randomized block design (CRBD) with replications as blocks; with only one replication grown at Romney in 2020. The four row plots were 1.83 m in length and 0.76 m spacing among rows, at a density of approximately 36 plants m$^{-2}$. In 2020, GDM seeds in Gibson, IL, planted all accessions in two row plots of 5 m long and 0.72 m spacing among rows with three replications in a CRBD, at a density of approximately 38 plants m$^{-2}$. Geographical coordinates of the four experimental locations are: 40° 28' 20.5" N and -86° 59' 32.3" W (Acre-IN), 40° 14' 09.2" N and -86° 53' 24.5" W (Romney-IN), 40° 28' 22.944" N and -88° 20' 43.152" W (Gibson- IL); 38° 54' 50.63" N and -92° 17' 33.67221" (Columbia-MO). S1 Fig shows a map of the experimental locations and delimitating MG III and IV zones that we elaborated using the R package "ggplot2" [60]. Combination of years and locations as unique environments resulted in nine environments. S1 Table presents the planting dates for individual environments.

We observed the plots for R1, R8, and RL in all environments; R1 was recorded as the number of days from planting until 50% of the plants in a plot had one open flower at any node on

the main stem, R8 was defined as the number of days after planting when 95% of pods had reached maturity color, and RL was calculated as the difference between R8 and R1 [61]. We visited the locations for ground visual dating of R1 as soon as the earliest flowering plots in each location began to show flowers and for R8 as soon as the earliest maturing plots in a given location began to senesce. Visits were every two to three days, and we interpolated dates when it was clear a plot reached R8 or R1 between visits. We expressed the date of plot flowering and maturity as the number of days after planting (DAP).

## Genotypic data, quality control and linkage disequilibrium estimation

The complete USDA soybean germplasm collection has genotype data of 42,509 SNPs (http://www.soybase.org/dlpages/#snp50k) obtained by genotyping with the Illumina Infinium SoySNP50K iSelect Bead chip [62]. Our colleagues [63] retrieved and made available the genotypic data from the SoyBase website [64], coded as {0,1,2}, imputed for missing data, and cleaned for redundant SNPs and SNPs with MAF≤ 0.05. The coded allelic genotype as {0,1,2} corresponded to {AA, Aa, aa}; where AA is homozygous towards the reference genome Williams82 [63]. We performed principal component analysis (PCA) of whole-genome SNPs using R software and plotted the first two principal components (PC) for visualization (Fig 1). We obtained the genomic relationship matrix (GRM) by using the R package NAM and visualized it with a Heatmap (S2 Fig) [65].

To characterize the mapping resolution, the average extent of genome-wide LD between pairwise SNPs was estimated by euchromatin and heterochromatin regions. First, we calculated the squared correlation coefficient ($r^2$) of alleles between markers using the LD function in the R package NAM [65], which phases molecular markers using the expectation-maximization algorithm [66]. Due to differences in recombination rates between euchromatic and heterochromatic regions, we calculated $r^2$ separately for the two chromosomal regions; We defined the approximate physical length and positions of heterochromatic and euchromatic regions in the Wm82.a2.v1 as described previously [67]. To measure the extent of LD, we only used the $r^2$ between SNPs with pairwise physical distances smaller than 10 Mb in either the euchromatic or heterochromatic region of each chromosome. We computed the mean $r^2$ within intervals of 100 kb and drew the LD decay plots (S3 Fig); then, we calculated the LD decay rate by chromosome as the distance where the average $r^2$ dropped to half its maximum value [68].

## Data and association analysis

To minimize the effect of environmental variation, we estimated the genetic values as the Best Linear Unbiased Estimator (BLUE) for each line by a mixed model analysis using the R package 'lme4' [69]. We combined year and location factors into one environment term (Eq 1) (S1 Table). The linear model used to model genetic values was:

$$y_{ijk} = \mu + f(x) + \alpha_i + \beta_j + \gamma_k + e_{ijk} \tag{1}$$

Where $y_{ijk}$ is the phenotype (R1, R8, RL) measured in the $j$th environment into the $k$th block, $\mu$ is the intercept, $f(x)$ is the spatial covariate based on a moving-average of neighbor plots as described by [70] and estimated through functions NNscr/NNcov of R package NAM [65], $\alpha_i$ captures the fixed genotype effects, $\beta_j$ ($j = 1,\ldots$number of environments) is the random environment effect with $\beta_j \sim N(0, \sigma_\beta^2)$ where $\sigma_\beta^2$ is the environment variance, $\gamma_k$ is the random effect of the $k$th block with $\gamma_{k(j)} \sim N(0, \sigma_\gamma^2)$ where $\sigma_\gamma^2$ is the block variance, and $e$ is the residual term distributed as $e_{ijk} \sim N(0, \sigma_e^2)$ where $\sigma_e^2$ is the residual variance. We estimated the BLUEs

for individual years, environments, and across all years and environments by modifying Eq 1. To estimate broad sense heritability (H) on an entry-mean basis, we used the same model structure from Eq 1 with only the spatial covariate treated as fixed effect; while, the accessions were treated as random effect and assumed to be normally distributed as $\alpha_i \sim N(0, \sigma_\alpha^2)$ where $\sigma_\alpha^2$ is the genetic variance. We estimated H from the REML variance components as:

$$H = \frac{\sigma_\alpha^2}{\sigma_\alpha^2 + \frac{\sigma_e^2}{r}} \tag{2}$$

Where $H$ corresponds to broad sense heritability in mean entry bases, $\sigma_\alpha^2$ is genetic variance, $\sigma_e^2$ corresponds to variance of error, and r is the number of replications. To estimate narrow-sense heritability ($h^2$) we used a whole-genome regression using the expectation-maximization restricted maximum likelihood method from the "NAM" package [65]. In this model the accessions BLUPs were modeled as a function of the polygenic effects $g \sim N(0, \mathbf{K}\sigma_g^2)$, where $\sigma_g^2$ is the additive genetic variance and $\mathbf{K}$ is a matrix of kinship coefficients estimated by genomic data, and the residual term $e \sim N(0, \sigma_e^2)$ where $\sigma_e^2$ is the residual variance. We estimated $h^2$ from the variance components as:

$$h^2 = \frac{\sigma_g^2}{\sigma_g^2 + \sigma_e^2} \tag{3}$$

To conduct the GWAS, we used an empirical Bayesian framework using the R package NAM, which uses a sliding-window strategy to increase power and avoid double fitting markers into the model [65]. For each trait, we used the BLUEs from Eq 1 as phenotypes. The model includes a polygenic term that accounts for the population structure to minimize false positives and increase statistical power. We tested marker-trait associations using a linear model implemented in the function *gwas*2:

$$\mathbf{y} = \mu + \mathbf{Z}\boldsymbol{u} + \mathbf{g} + \boldsymbol{e} \tag{4}$$

Where the BLUEs (**y**) obtained in Eq 1 were modeled as a function of an intercept (μ), the incidence matrix (**Z**) relating the accessions to the marker effects $\boldsymbol{u} \sim N(0, \mathbf{I}\sigma_u^2)$, the vector of independent residuals $\boldsymbol{e} \sim N(0, \mathbf{I}\sigma_e^2)$, and the polygenic effect $\boldsymbol{g} \sim N(0, \mathbf{K}\sigma_g^2)$ accounting for the genetic covariance among individual through the genomic relationship matrix **K**. Statistical significance of markers was measured by calculating the likelihood ratio test statistics (LRT), which represent the improvement that each SNP provides to a mixed model when compared to the reduced model without the marker. The p-values were obtained from LRT using the Chi-squared density function with 0.5 degrees of freedom [65]. We used the multiple testing correction developed by [71] to evaluate marker trait associations. The corrected p-value threshold (0.000068) was obtained by applying a Bonferroni correction to the significance level (p-value 0.01) using the number of independent tests (143). The number of independent tests corresponded to the number of principal components accounting for a large portion of the genotypic variance (95%). We used the R package CM-plot to create the Manhattan plots [72] and the R package "ggplot2" to generate the plots of the genomic regions with the stronger signals [60].

We further inspected seven association signals to identify their potential functional effects. Based on Wm82.a2.v1 genome assembly, selected SNPs are positioned in Chr 4 at 17,075,267 (G/A) bp, 17,228,343 (T/C) bp, 40,009,617 (C/T) bp, 40,276,263 (A/G), 40,151,473 (T/C) bp and 40,218,961 (G/A) bp and in Chr 10 at 41,455,680 (G/A) bp (Table 1). In GWAS, the strongest associated variants are rarely the causal mutation (CM); instead, they are likely in linkage

**Table 1. Summary of single-nucleotide polymorphisms (SNPs) significantly associated with flowering time (R1), maturity time (R8), and reproductive length (RL) in three hundred twenty-nine _G. max_ accessions across environments and years.**

| Chr | SNP Wm82.a2[a] | SoySNP50k_ID [a] | Trait | Peak | -log(p-values) | Effect | Reported QTLs[a] |
|-----|----------------|------------------|-------|------|----------------|--------|------------------|
| **10** | Gm10_41455680_G_A* | ss715607080 | R1 | Peak-3 | 4.4 | 1.79 | R8 full maturity 10-g4.1, Reproductive stage length 4-g2.1, Reproductive stage length 4-g2.2, Reproductive period 3-g5, Seed protein 7-g7 and Seed weight 4-g10 |
| **4** | Gm04_17228343_T_C* | ss715587206 | R8 | Peak-1 | 4.71 | -1.85 | - |
| **4** | Gm04_17075267_G_A* | ss715587203 | R8 | Peak-1 | 4.4 | -1.77 | - |
| **4** | Gm04_40009617_C_T* | ss715587845 | R8 | Peak-2 | 4.66 | -1.77 | Reproductive stage length 2–2, 3–4, and 5–1, Pod maturity 18–3[b] |
| **4** | Gm04_40276263_A_G* | ss715587857 | R8 | Peak-2 | 4.7 | -1.76 | [b] |
| **4** | Gm04_16673792_G_A | ss715587192 | R8 | Peak-1 | 4.4 | -1.77 | - |
| **4** | Gm04_16889396_T_C | ss715587197 | R8 | Peak-1 | 4.4 | -1.77 | - |
| **20** | Gm20_46615517_A_C | ss715638773 | R8 | Peak-4 | 4.33 | -1.36 | First flower 6-g4, R8 full maturity 8-g13, Reproductive period 3-g1 |
| **4** | Gm04_39977826_G_A | ss715587844 | R8 | Peak-2 | 4.17 | -1.6 | [b] |
| **4** | Gm04_40151473_T_C* | ss715587852 | R8 | Peak-2 | 4.18 | -1.68 | [b] |
| **4** | Gm04_40218961_G_A* | ss715587856 | R8 | Peak-2 | 4.18 | -1.68 | [b] |
| **3** | Gm03_36427644_C_T | ss715585727 | RL | Peak-5 | 4.17 | -1.62 | First flower 4-g10, First flower 3-g2, R8 full maturity 3-g3 |
| **4** | Gm04_40151473_T_C* | ss715587852 | RL | Peak-2 | 5.63 | -1.92 | [b] |
| **4** | Gm04_40218961_G_A* | ss715587856 | RL | Peak-2 | 5.63 | -1.92 | [b] |
| **4** | Gm04_17228343_T_C* | ss715587206 | RL | Peak-1 | 4.24 | -1.68 | - |
| **4** | Gm04_40009617_C_T* | ss715587845 | RL | Peak-2 | 4.35 | -1.65 | [b] |
| **4** | Gm04_39006019_T_C | ss715587808 | RL | Peak-2 | 4.22 | -1.58 | [b] |
| **4** | Gm04_39484148_T_C | ss715587823 | RL | Peak-2 | 4.22 | -1.58 | [b] |
| **4** | Gm04_39731223_A_G | ss715587834 | RL | Peak-2 | 4.22 | -1.58 | [b] |

SNP Wm82.a2 is the SNP positions based on Wm82.a2

[a] Information obtained from SoyBase and scientific reports

[b] the same reported QTLs applied for all SNPs withing peak 2 in Chr4

* Positions of the tagging SNPs used for candidate gene selection. SNP IDs, location in the genome, variance explained, distance from known _E_ genes and chromosomal region are presented in an expanded version of this table in S6 Table

disequilibrium (LD) with the causal variant, especially when using low-density genotype data [68, 73].

We followed three steps to identify potential causal genes: First, we identified an initial list of potential causal genes by using the Synthetic Phenotype to CM strategy (SP2CM) developed by our colleagues [73] and implemented in AccuTool (https://soykb.org/AccuTool/index.

php). In SP2CM and AccuTool, the positions of the SNPs significantly associated with the traits were treated as synthetic phenotypes, and their correspondence with other genomic variants within a specified region was measured through an average accuracy estimate, whose equations are described by [73]. An average accuracy of 100% indicates an exact match between the significant SNP and the other genomic variant in the Soy775 accession panel. The Soy775 accession panel combines all publicly available resequencing data from 775 soybean accessions, which increases AccuTool's power of detection [73]. We explored a window of +/- 1 Mb distance from the significant SNPs. Genes with genomic variants that are in high correspondence (average accuracy >85%) with significant SNPs were selected as potential candidate genes. According to our colleagues [73] using a correspondence threshold of 85% and a large interval reduces the risk of missing true positive associations without negative consequences on the analysis as any variant outside the range of LD will show low correspondence with the significant SNPs. AccuTool uses SNP positions and annotated genes from the Williams 82 a2. v1 reference genome, whose source is SoyBase (www.soybase.org). Second, we retrieved the previously characterized functions of the list of potential causal genes generated in the first step. To determine the gene's functions, we searched SoyBase, NCBI RefSeq, UniProt, The *Arabidopsis* Information Resource (TAIR), and scientific articles. Lastly, we selected a final list of genes that meet at least one of the two following criteria: 1) genes have a known function in soybean related to the studied traits; 2) genes have orthologs in *Arabidopsis* or other plants, whose functions are related to the studied traits.

## Results

### Phenotypic variation and genetic diversity

Despite sharing the same haplotype for major *E* genes (*e1-as/E2/E3*), the 329 accessions exhibited a wide range of variation in R1, R8, and RL. The phenotypic distribution of R1, R8, and RL by environment is presented in S4 Fig. Across environments, R1 ranged from 31 to 80 DAP, R8 ranged from 110 to 169 DAP, and RL ranged from 55 to 111 days. In addition, R8 and RL were the least variable traits, whereas R1 was the most variable. Descriptive statistics by accessions, environments, and across all environments are provided in S2 and S3 Tables, alongside broad- and narrow-sense heritability. From all environments, Gibson, IL in 2020 (E9) had the highest mean values and widest ranges for R1, R8, and RL; and it was planted earlier compared to the other environments (S1 Table). The observed broad sense heritabilities were 66% for R8, 52% for RL and 61% for R1, while narrow-sense heritabilities were 69% for R8, 57% for RL, and 65% for R1.

   To investigate the population structure of the selected 329 accessions, we generated a scatter plot of the first two PCs from PCA of whole-genome SNPs in the 329 accessions (Fig 1), that in combination with k-mean clustering (S5 Fig) indicated the existence of two subgroups in the studied panel of accessions. The GWAS model included a polygenic term that accounts for the population structure to minimize false positives and increase statistical power. A total of 31,689 polymorphic SNPs with MAF $\geq$ 0.05 were used for the association analysis with an average marker density of 1 SNP every 30 kb genome-wide, and ranging from 21 kb/SNP (Chr 13) to 44 kb/SNP (Chr 1) (S6 Fig). The majority of the SNPs (76%) were located in euchromatic regions (S4 Table). Due to the differences in LD decay rates between chromosomal regions, a higher marker density is required for the low LD euchromatic regions, whereas in the heterochromatic region a lower marker density is acceptable [41, 74]. The average LD decay rate of the panel of accessions in euchromatic regions was 2150 kb, with $r^2 = 0.38$ being half of its maximum value; while, in heterochromatic regions, the $r^2$ dropped to half of its maximum until 8650 kb (S5 Table, and S3 Fig). Given the average marker density of 30 kb/SNP

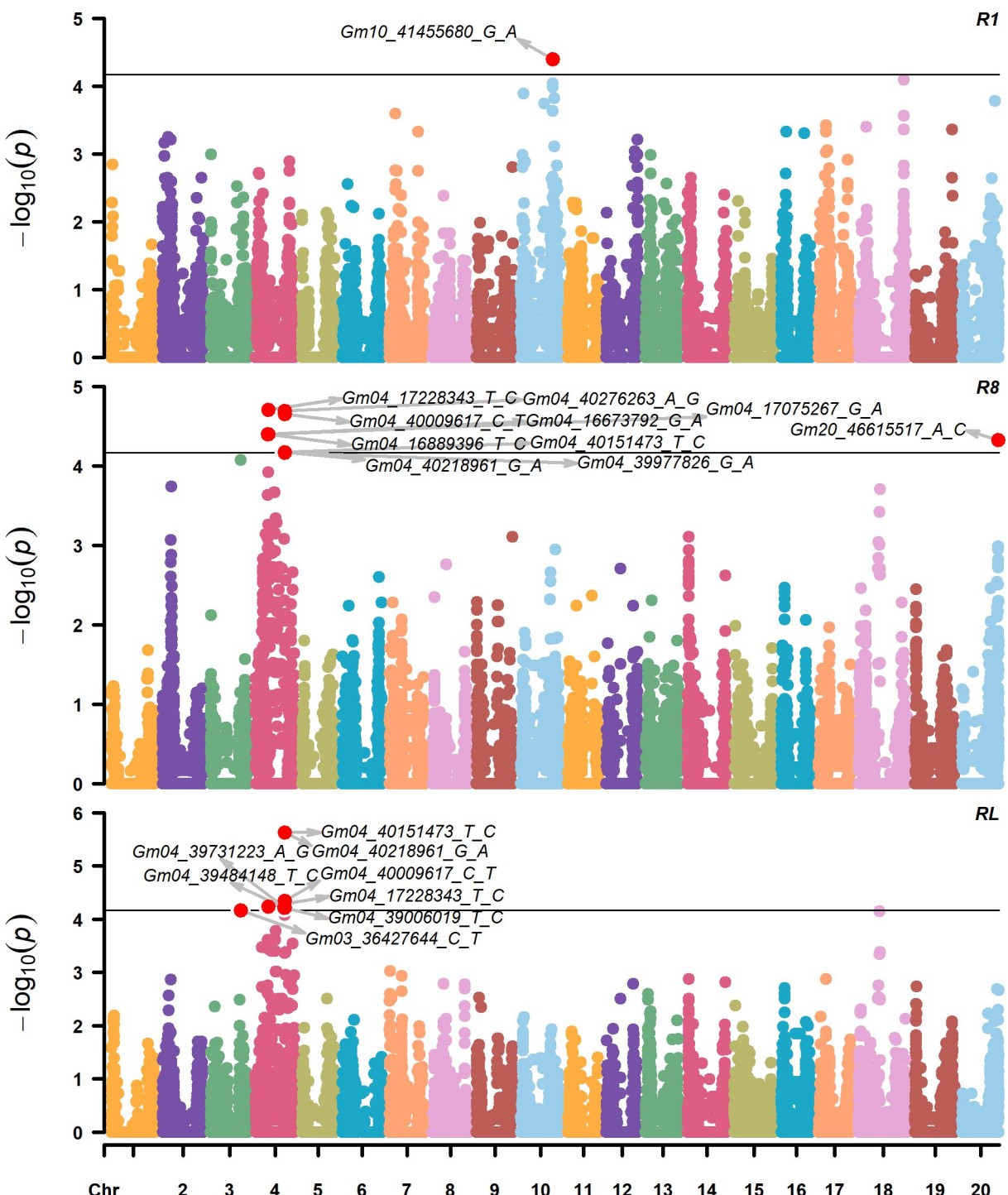

**Fig 2. Manhattan plots of GWAS for flowering time (R1), maturity time (R8), and reproductive length (RL) in three hundred twenty-nine *G. max* accessions.** GWAS results correspond to the analysis across the nine environments. The horizontal dashed lines indicate the statistically significant cut-off of–log (p-value) = 4.16. Significant SNPs IDs correspond to the Wm82.a1.

and the LD decay rate, the used SNPs were expected to have reasonable power to identify loci significantly associated to the studied traits.

## Genome-wide association analysis

Successful identification of marker-trait associations and the power of resolution relies on the size and variability of the mapping population [75]. In soybeans, as few as 9,600 SNPs can capture most of the haplotype variation [75, 76]; thus, this study used a very diverse population and a sufficient number of SNPs (Fig 1 and S6 Fig). GWAS across environments revealed a total of 15 SNPs associated with at least one of the phenotypes with $p$-values $< 6.85$ x $10^{-5}$ (Fig 2). One SNP was associated with R1, ten SNPs with R8, eight SNPs with RL, and four SNPs with R8 and RL. Of the ten SNPs associated with R8, nine were from Chr 4 with two strong peaks, and one was from Chr 20. One SNP associated with RL was from Chr 3, and the other seven SNPs were from Chr 4, with the same two strong peaks observed for R8. The shared significant SNPs between R8 and RL were from Chr 4. The effect of significant SNPs on trait phenotypes ranged from -1.38 to 1.92 DAP. A summary of the association signals with the effect of each SNP on days to R1 and R8, and RL is presented in Table 1 and, an expanded version of the table is presented in S6 Table, including additional information about the significant SNPs such as variance explained, chromosomal region, and SNP ID.

GWAS, by unique environments and years, revealed a total of 5 SNPs associated with R1, 14 SNPs associated with R8, and 13 SNPs associated with RL; ten SNPs were associated with R8 and RL. From them, we identified four, six, and five environment-specific association signals for R1, R8, and RL, respectively. Environment-specific SNPs may be susceptible to environmental influences and could play an essential role in soybean adaptation to specific environments. A summary of the association signals from the different environments and years, alongside the effect and variance explained by each SNP marker for R1, R8, and RL is presented in S7 Table, and S7 Fig. We selected the six most significant SNPs from Chr4 and one from Chr 10 to examine further their surrounding genomic region for genes that can potentially explain their functional effect on the studied traits. Selected SNPs from Chr 4 were significantly associated with R8 and RL across all environments, had the smallest P-values and the highest effects, and were present in more than one environment (S6 and S7 Tables); in addition, we included the SNP on Chr 10 because it was the only significant association observed for R1 across all environments. In this paper, we referred to the selected SNPs as tagging SNPs (Table 1, and S6 Table). To provide further verification of the tagging SNPs, we examined their effect on trait's performance and observed significant differences between accessions carrying the reference and alternative alleles for all SNPs (S8 Fig).

## Candidate gene exploration

We explored a genomic region of +/- 1 Mb surrounding the seven tagging SNPs to gain insight into their potential functional effects. Local Manhattan plots, linkage disequilibrium analysis (LD), and SNPs physical positions revealed that the tagging SNPs are likely representing three different association peaks, one on Chr 10 associated with R1 and two on Chr 4 associated with R8 and RL (Figs 3, 4, and S5 Table). We defined the regions of the peaks taking the two most distant significant SNPs of each peak. Based on the Williams 82 a2.v1 reference genome, one peak was on Chr 4 at 16,673,792–17,228,343 bp (peak-1), a second peak was also on Chr 4 but at 39,006,019–40,276,263 bp (peak-2), and a third peak was on Chr 10 at 41,455,680 (peak-3). Of the seven tagging SNPs, two were from peak-1, four from peak-2, and one from peak-3 (Table 1). Although the peak-3 on Chr 10 had only one significant SNP, we decided to examine it further since it was the only significant SNP for R1 and had an effect of +/- 2 DAP on

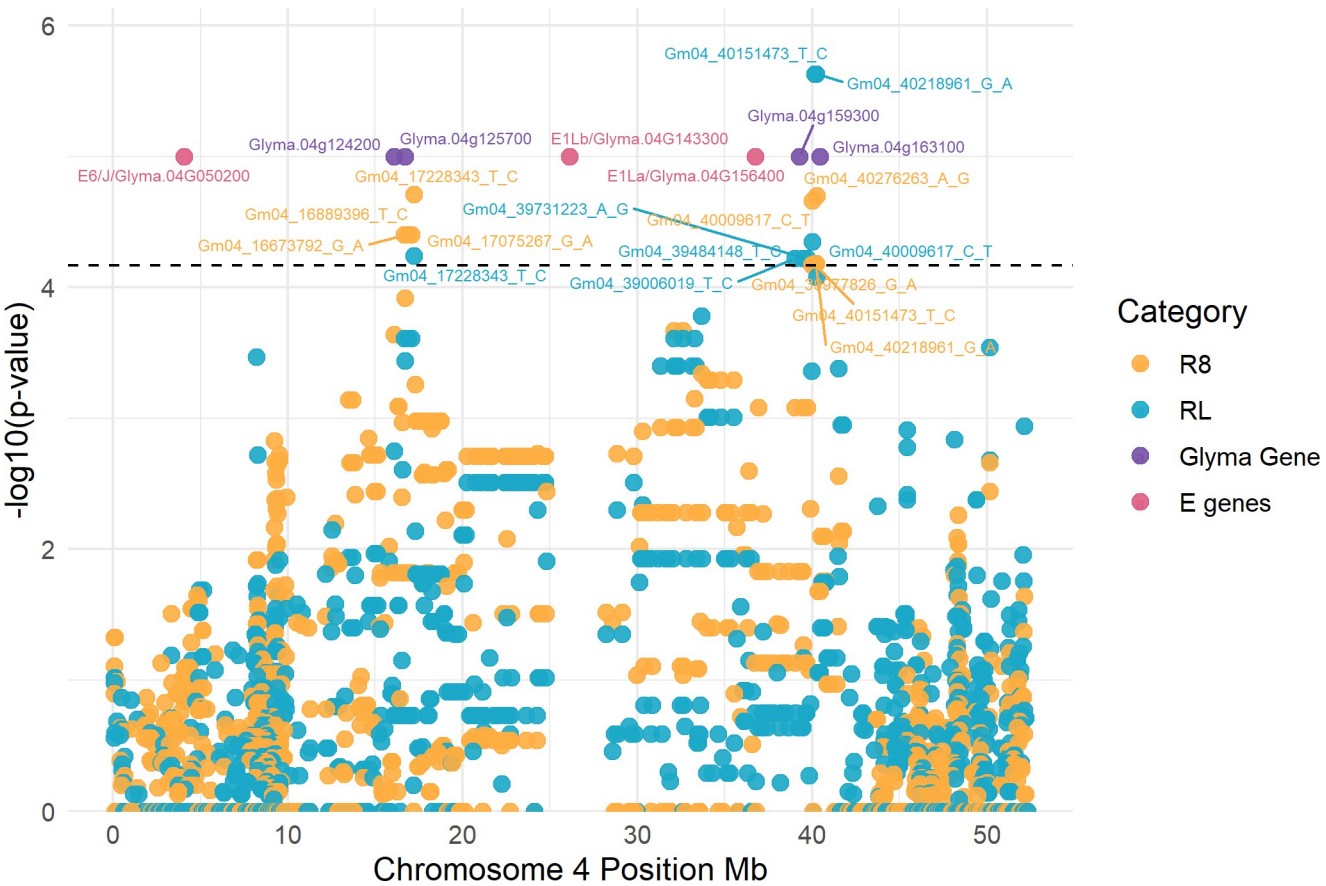

**Fig 3.** GWAS results showing the candidate genomic regions of peak-1 (left) and peak-2 (right) for R8 (yellow) and RL (blue). In the graph there are labels for the significant SNPs, the known maturity *E* genes (pink), and the potential causal genes in chromosome 4 (purple) which were obtained using AccuTool. The horizontal dashed lines indicate the statistically significant cut-off of –log (p-value) = 4.16. Significant SNPs IDs correspond to the Wm82.a1.

flowering time (Table 1, and S6 Table). There were four previously reported QTLs, three for RL and one for R8 in the genomic region of peak-2 on Chr 4 [43, 57, 77, 78]. Similarly, there were four previously reported QTLs in peak-3 on Chr 10, one for R8 and three for RL according to the SoyBase.org browser and previous studies (Table 1) [79, 80]. Significant peaks on Chr 4 are at least 2 and 9 Mb apart from maturity genes *E1La*, *E1Lb*, respectively, whereas the significant SNP on Chr 10 is 4 Mb away from maturity gene *E2* (Table 1, and Figs 3 and 4).

To explore the genomic region of +/- 1 Mb around the tagging SNPs, we used the SP2CM implemented in AccuTool [73], which measures the correspondence of the tagging SNPs with other genomic variants within the specified region through an average accuracy estimate. This methodology allows the assessment of the relationship between tagging SNPs and variant alleles of nearby genes. Variant alleles in a total of 53 *Glyma* genes showed correspon-dence (average accuracy > 85%) with the tagging SNPs of peak-1, variant alleles of 46 *Glyma* genes had high correspondence with the tagging SNPs of peak-2, and variant alleles of 12 *Glyma* genes had high correspondence with the tagging SNPs of peak-3. We designated six genes as potential candidates, with two genes per peak. The selected genes are interesting because they had orthologs in *Arabidopsis* and other plant species with functions related to flowering and maturity, even though their exact functions in soybeans are unknown (Table 2). For each gene, we selected the variant allele with the highest correspondence with tagging SNPs as the most likely cause of the effect on trait phenotypes (Table 2). From them, one was a

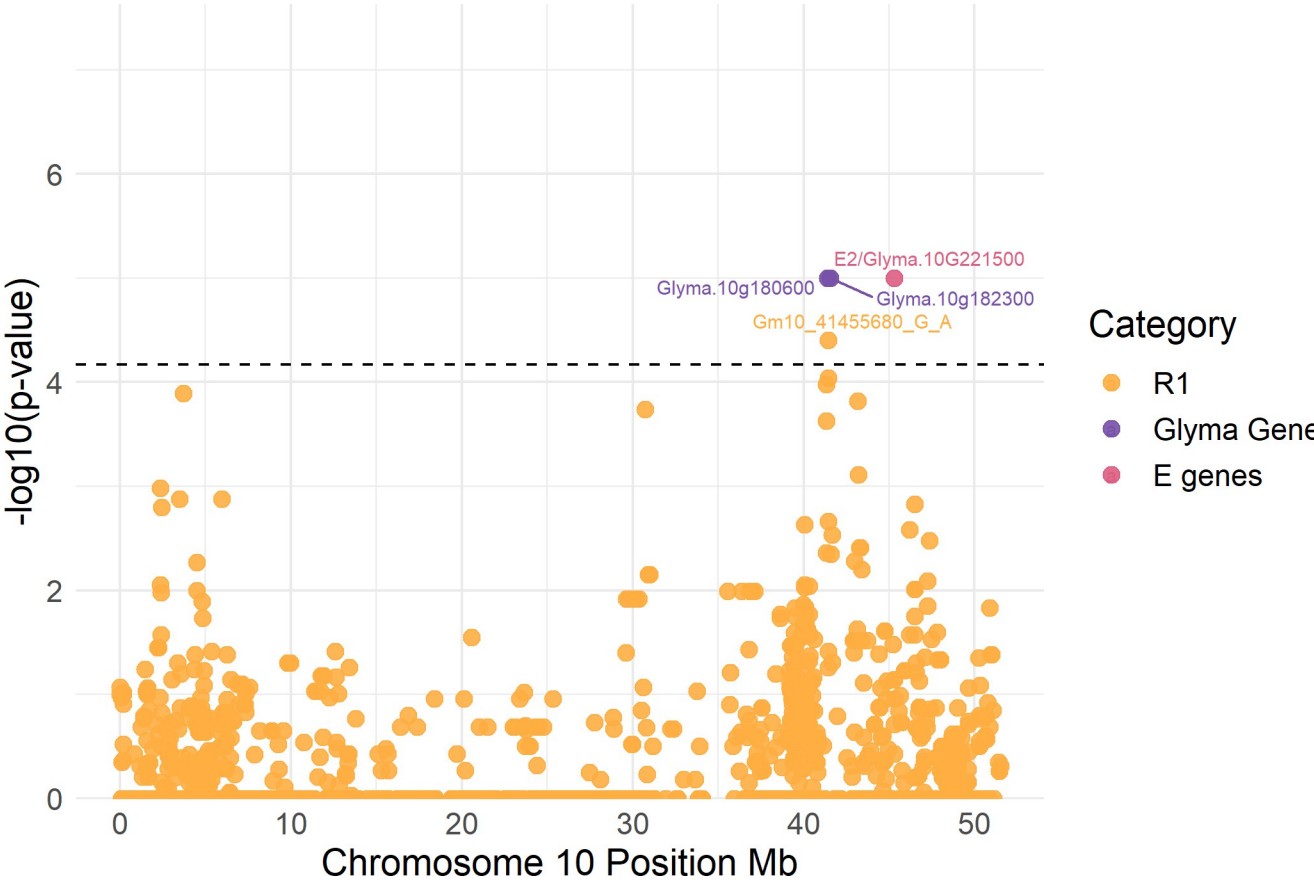

**Fig 4. GWAS results showing the candidate genomic region of peak-3 for R1.** In the graph there are labels for the significant SNPs, the known maturity E genes (pink), and the potential causal genes in chromosome 10 (Purple), which we obtained using AccuTool. The horizontal dashed lines indicate the statistically significant cut-off of–log (p-value) = 4.16. Significant SNPs IDs correspond to the Wm82.a1.

modifying variant at nucleotide 113 (A to T) in *GmFulb* (*Glyma.04G159300*) which caused an amino acid change (H to L) [77]; the gene mentioned above is the most interesting candidate of chromosome 4. Local Manhattan plots of the genomic regions showing the association signals on Chr 4 and 10, nearby known *E* genes, significant SNPs, and potential candidate genes are presented in Figs 3 and 4. A complete list of genes that had high correspondence with the different tagging SNPs is presented in S8 Table.

Using the Soybean Allele Catalog Tool (https://soykb.org/SoybeanAlleleCatalogTool/), we identified 14 accessions from the studied soybean panel with sequence data available. In the 14 accessions, we inspected their allelic variation for the seven tagging SNPs, SNPs on the known *E* genes near the peaks, and allelic variants of the six candidate genes. Accessions showed allelic variation for the allelic variant (H38L) of candidate gene *GmFulb* at 04:39294836, while the allelic variants from the other five candidate genes are not reported in the Soybean Allele Catalog Tool because they are not modifying variants. Accessions also exhibited allelic variation for the six tagging SNPs from Chr 4 (Table 3). We compared the mean performance of lines carrying the reference and alternative alleles of tagging SNPs and the allelic variant of *GmFulb* that were polymorphic in the 14 accessions. There were significant differences in R8 and RL between accessions carrying the reference and alternative alleles for the allelic variant in the candidate gene *GmFulb* and the tagging SNPs from peaks on Chr 4. Two tagging SNPs from peak 2 on Chr 4 did not show significant different in RL among the 14 accessions (Fig 5). It is

**Table 2. Description of the candidate genes.**

| Gene ID | Chr | Variant[a] | Variant Position[b] | SNP Wm82.a2[b] | Peak | Avg Acc %[c] | Distance (Kb)[d] | Description[e] |
|---|---|---|---|---|---|---|---|---|
| Glyma.04g124200 | 4 | AT\|upstream_gene_variant | 16087021 | Gm04_17228343_T_C* | 1 | 91.1 | 1141.32 | BZIP transcription factor/ABA-responsive element binding protein 3 |
| | | | | Gm04_16673792_G_A | | 87.8 | 586.77 | |
| | | | | Gm04_16889396_T_C | | 87.2 | 802.38 | |
| | | | | Gm04_17075267_G_A* | | 86.8 | 988.25 | |
| Glyma.04g125700 | 4 | C\|upstream_gene_variant | 16707992 | Gm04_17075267_G_A* | 1 | 96.7 | 367.27 | Myb domain protein 33 |
| | | | | Gm04_17228343_T_C* | | 93.3 | 520.351 | |
| | | | | Gm04_16673792_G_A | | 100 | 34.2 | |
| | | | | Gm04_16889396_T_C | | 98.2 | 181.40 | |
| Glyma.04g159300 | 4 | A\|missense_variant\|H38L | 39294836 | Gm04_40218961_G_A* | 2 | 90.2 | 924.12 | [f]*GmFulb* an *AP1/FUL*-like gene that are mainly involved in reproductive transition, floral meristem identity, and fruit development in plants |
| | | | | Gm04_40151473_T_C* | | 89.3 | 856.64 | |
| | | | | Gm04_39006019_T_C | | 97.7 | 288.82 | |
| | | | | Gm04_39484148_T_C | | 99.6 | 189.31 | |
| | | | | Gm04_39731223_A_G | | 99.6 | 436.39 | |
| Glyma.04g163100 | 4 | G\|intron_variant | 40436493 | Gm04_40218961_G_A* | 2 | 93.9 | 217.53 | PHD finger-containing protein. Interacts with BDT1, acts with other PHD proteins to associate with flowering genes and thereby suppress their transcription. |
| | | | | Gm04_40276263_A_G* | | 97.3 | 160.23 | |
| | | | | Gm04_40151473_T_C* | | 92.5 | 285.02 | |
| | | | | Gm04_39484148_T_C | | 87.7 | 925.34 | |
| | | | | Gm04_39731223_A_G | | 87.7 | 705.27 | |
| Glyma.10g180600 | 10 | C\|upstream_gene_variant | 41423808 | Gm10_41455680_G_A* | 3 | 96.7 | 1019.19 | GmCRY2/Cryptochrome 2 |
| Glyma.10g182300 | 10 | G\|downstream_gene_variant | 41571115 | Gm10_41455680_G_A* | 3 | 96.4 | 1019.19 | COP1-interacting protein-related |

[a] if a tagging SNP had high correspondence with more than one variant within a gene, we selected the variant with the highest average accuracy

[b] SNPs and candidate gene variants positions based on Glyma.Wm82.a2.v1 genome assembly

[c] Average accuracies obtained by AccuTool among gene variants and significant SNPs, we only reported the accuracies >85%

[d] Distance of the candidate gene variants from the significant SNPs

[e] Description and gene name of orthologous genes in *Arabidopsis*. [e] Function reported for Soybean by Jia et al. 2015

* Positions of the tagging SNPs used for candidate gene selection

important to note that the set of 14 sequenced accessions did not show allelic variation for *E1la* and *E1lb* genes, the tagging SNP on Chr 10, or maturity gene *E2*. The above results support *GmFulb* as the most likely candidate for one of the peaks on Chr 4.

## Discussion

Despite sharing the same *e1-as*, *E2*, and *E3* alleles, wide ranges of variation and moderate to high broad sense heritabilities were observed in the studied traits, which reflects the complexity of the genetic control underlying R1, R8, and RL in soybeans. A previous study also observed wide ranges of variation, and moderate to high heritabilities for R1 and R8 in a mapping population with the same *e1-asE2E3* haplotype for major *E* genes [57]. Of the three phenotypes, R1 and RL were under a more substantial genetic control than R8 (S3 Table); however, heritability estimations depend in great part on the properties of the studied individuals and the environments with different heritabilities reported for R1, R8, and RL among previous studies [81, 82]. It is important to note that the effect of environmental factors on R1, R8, and RL is not well documented. Future studies that better understand environmental effects on *E* genes would help elucidate the mechanisms regulating flowering time and facilitating site-specific soybean breeding.

**Table 3. Allelic variants of tagging SNPs, and candidate gene *GmFulb* (*Glyma.04159300*) for 14 accessions.**

| Accession | Tagging SNPs[a] | | | | | | Glyma.04g159300 [b] |
|---|---|---|---|---|---|---|---|
| | Gm04 40009617 | Gm04 40151473 | Gm04 40218961 | Gm04 40276263 | Gm04 17075267 | Gm04 17228343 | 39294836 |
| PI515961 | C\|Ref | T\|Ref | G\|Ref | A\|Ref | G\|Ref | T\|Ref | T\|Ref |
| PI547501 | C\|Ref | T\|Ref | G\|Ref | A\|Ref | G\|Ref | T\|Ref | T\|Ref |
| PI547860 | C\|Ref | T\|Ref | G\|Ref | A\|Ref | G\|Ref | T\|Ref | T\|Ref |
| PI547885 | C\|Ref | T\|Ref | G\|Ref | A\|Ref | G\|Ref | T\|Ref | T\|Ref |
| PI548603 | C\|Ref | T\|Ref | G\|Ref | A\|Ref | G\|Ref | T\|Ref | T\|Ref |
| PI591495 | C\|Ref | T\|Ref | G\|Ref | A\|Ref | G\|Ref | T\|Ref | T\|Ref |
| PI634761 | C\|Ref | T\|Ref | G\|Ref | A\|Ref | G\|Ref | T\|Ref | T\|Ref |
| PI548543 | T\|Alt | T\|Ref | G\|Ref | G\|Alt | G\|Ref | T\|Ref | T\|Ref |
| PI548634 | T\|Alt | T\|Ref | G\|Ref | G\|Alt | G\|Ref | T\|Ref | T\|Ref |
| PI597382 | T\|Alt | T\|Ref | G\|Ref | G\|Alt | G\|Ref | T\|Ref | T\|Ref |
| PI547811 | T\|Alt | C\|Alt | A\|Alt | G\|Alt | A\|Alt | C\|Alt | A\|H38L |
| PI548177 | T\|Alt | C\|Alt | A\|Alt | G\|Alt | A\|Alt | C\|Alt | A\|H38L |
| PI548180 | T\|Alt | C\|Alt | A\|Alt | G\|Alt | A\|Alt | C\|Alt | A\|H38L |
| PI548632 | T\|Alt | C\|Alt | A\|Alt | G\|Alt | A\|Alt | C\|Alt | A\|H38L |

[a]SNP IDs and allelic information obtained from the genotype data used
in this study

[b] allelic information obtained from the Soybean Allele Catalog Tool.

We observed higher LD decay rates compared to previous reports for accessions of the USDA soybean germplasm collection; however, this could be due to differences in the structure and diversity of the specific set of the USDA germplasm collection used in each study [41, 63, 74, 83, 84]. LD estimates are highly dependent on the sample size, and small sample sizes tend to overestimate the amount of LD [85]; therefore, our LD decay rates for euchromatic and heterochromatic regions may be inflated by the smaller sample size, lower marker density in heterochromatic regions and the way we selected the soybean panel. LD is the primary factor limiting the mapping resolution in GWAS, and overestimation of the extent of LD could lead to misinterpretations of the levels of LD in the studied populations and the sizes of identified QTLs [41, 86]; therefore, we also considered previous reports on the levels of the LD in the USDA soybean germplasm collection as a reference [63, 83].

Previous research demonstrated that maturity loci *E1*, *E2*, and *E3* are significant determinants of soybean maturity and flowering under field conditions controlling up to 66% of the variation [23, 24, 44, 45]. Here, we identified additional loci with an effect of up to two days on flowering and maturity that explain less than 30% of the observed variation, suggesting the existence of other undetected minor effect loci controlling these traits and a strong influence of the environment. From the significant SNP-trait associations, two were stable across environments, while others were environment specific. Finding stable and real associations that can be detected in various environments and plant materials is one of the major difficulties of GWAS studies [80], mainly due to the strong effect that environmental factors have on complex traits such as R1, R8, and RL. Stable marker-trait associations are helpful for marker-assisted selection (MAS), which is generally the goal of GWAS [87]. In contrast, environment-specific SNP-trait associations may play an essential role in adaptation to changing environments and maintenance of genetic variation in populations [88]; these SNPs might enhance or diminish phenotype when MAS is applied in a specific environment. The two detected peaks

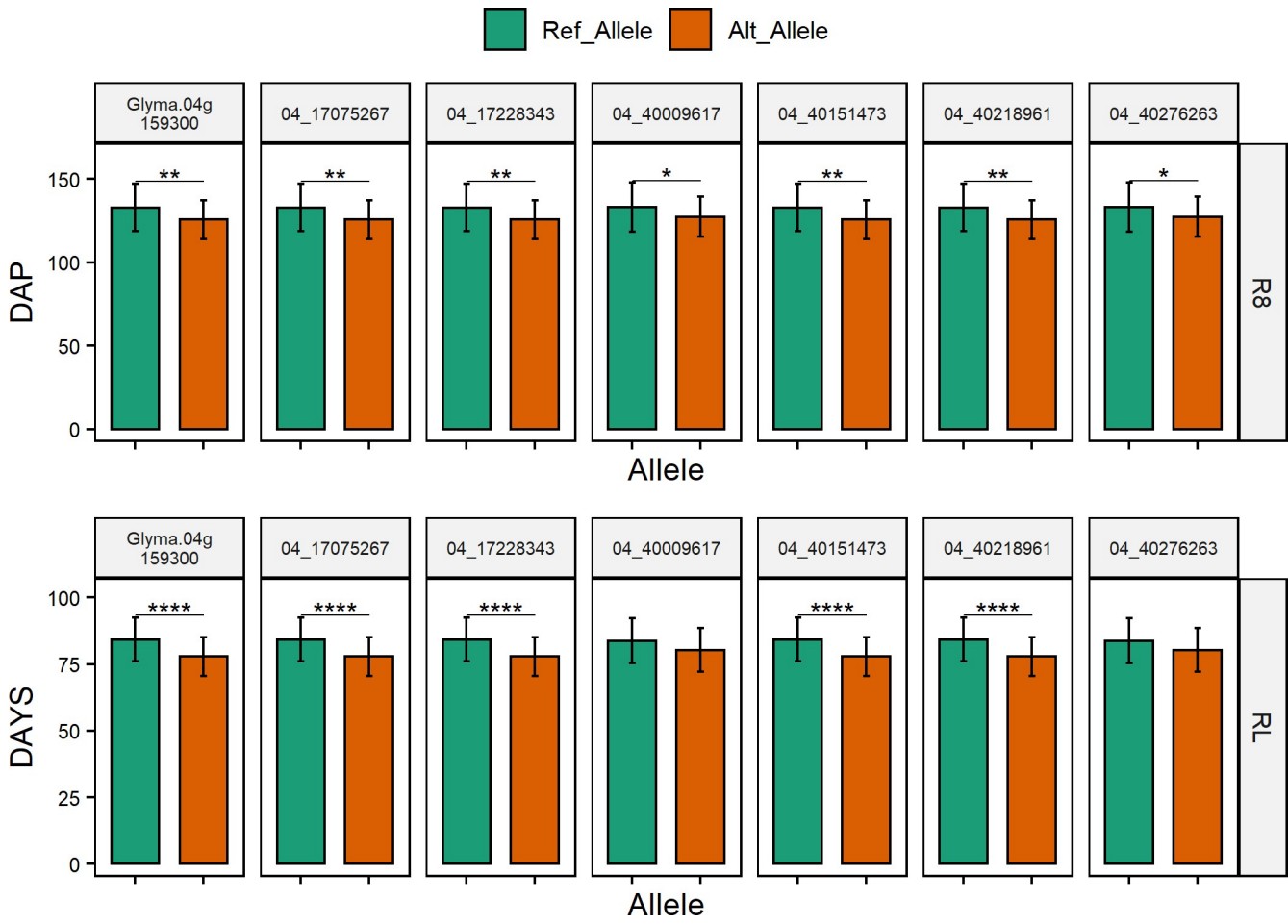

**Fig 5. Phenotypic differences between lines carrying the reference allele (Ref_Allele) and the alternative allele (Alt_Allele) of the SNP in the *GmFulb* gene, and six tagging SNPs associated with R8, and RL.** ** indicates significant differences at a p-value ≤ 0.01 between the two groups. **** indicates significant differences at a p-value ≤ 0.0001 between the two groups. DAP is days after planting.

on Chr 4 are good candidates for validation studies, which is necessary before subsequent research in QTL cloning or marker-assisted breeding for soybean yield improvement.

Previous research showed that QTLs associated with R8 and major maturity *E* genes also affected R1 and RL mainly because they are highly correlated traits [6, 29, 57, 79]; however, in this study, pleiotropic effects existed only over R8 and RL. Our results support the idea that *E* genes can influence R8 through their influence on RL; however, there is still limited knowledge of whether *E* genes control maturity by simply influencing flowering time, reproductive length, or both [11]. In addition to the maturity loci *E1* to *E3* fixed in the studied panel of accessions, we did not detect any of the other known maturity loci. No detection of other major *E* loci could result from the lack of functional polymorphisms at these loci in the studied soybean panel or the inability to detect genetic variants on those loci due to low SNP coverage. Based on our findings, trait-specific minor effect loci could also be involved in the control of R1 and R8 in soybeans, as suggested in a previous study [41]. In addition, earlier maturing varieties sharing the same genotype for major *E* loci may be selected independently of flowering time.

One of the two stable Chr4 peaks resides about 3.2 Mb from *E1La* and 19 Mb from *E1Lb*, and the other is 14 Mb apart from *E1La* and 9 Mb from *E1Lb*, whereas the significant SNP on

Chr 10 was located at 3.8 Mb apart from the *E2* gene [10, 18, 20, 22, 38, 39]. Significant Chr 4 peaks are located in heterochromatic regions, while the Chr 10 peak resides in an euchromatic region. In soybeans, LD decay rates reported for euchromatic regions are around 113 to 326 kb, while LD decay rates for heterochromatic regions range from 4 to 9 Mb [41, 63, 74, 83]. As a result, the significant SNP detected on Chr 10 does not likely correspond to the known *E2* locus. In contrast, the distance of *E1la* and *E1lb* from the two peaks on Chr 4 brings to question if the effect detected is caused by *E1la* and *E1lb*; however, there was no high correspondence between the tagging SNPs and *E1la* and *E1lb*. In addition, the absence of genetic variation for *E1la* and *E1lb* in the presence of genetic variation for the tagging SNPs suggested that detected peaks on Chr 4 in this study do not correspond to known *E* genes.

The significant peaks detected for R8 and RL on Chr 4 are located in a region similar to previously reported QTLs for R8 and RL [43, 57, 77, 78]. Researchers [57], using a bi-parental RIL population sharing the same *e1-asE2E3* haplotype for *E* genes, mapped a QTL encompassing the two Chr 4 peaks; thus, specific alleles of these major *E* genes may regulate significant peaks on Chr 4. It is important to note that GWAS using natural populations offers a higher mapping resolution than linkage-based analysis [89]. Here, we identified two association peaks 23 Mb apart from each other rather than a large single QTL as described by [57]. As opposed to [57], we did not observe an effect from Chr 4 peaks over R1; therefore, it may be possible that Chr 4 peaks are also affected by other maturity loci causing different effects on R1. In addition, a previous study found a major QTL on Chr 4 for R8 and RL that overlaps with peak-2 using a biparental RIL population, whose parents Noir (*e1-ase2e3fsE4*) and Archer (*e1-ase2E3E4*) had the same allele *e1-as* as our studied population [77]. Interestingly, they also reported *GmFulb* as the candidate gene of their QTL on Chr 4 [77], which differed in sequence between the parents of the RIL population with one of the parents harboring an amino acid variation (H38L) caused by a base substitution. This study also identified the allelic variant H38L within the *GmFulb* gene as the most likely contributor to Peak-2's effect on R8 and RL.

The most intriguing candidate for peak-2 on Chr 4 was *GmFulb* (*Glyma.04g159300*), part of the *APETALA1 (AP1)/ FRUITFULL (FUL)* genes of the MADS-box transcription factor superfamily [90]. Its orthologous genes in *Arabidopsis* and other plant species play important roles in specifying floral meristem identity, reproductive transition and determining the fate of floral organ primordia [91, 92]. In soybeans, a previous study found eight *AP1/FUL*-like genes, including our candidate gene *GmFULb* (*Glyma.04G159300*) [90]; where the homologous gene, *GmFULa*, has shown to play roles in flowering and maturation, and to improve soybean yield by enhancing carbon assimilation [90, 93, 94]. Furthermore, *GmFULb* was one of the *E1* downregulated genes identified when overexpressing *E1* [94]; therefore, the effect of Peak-2 over R8 and RL is likely dependent on *E1* gene activity. Our results and a previous study [77] suggested that the modifying variant SNP at position 04:39294836 in *GmFulb* (H38L) is the most relevant candidate for peak-2 influencing R8 and RL.

The other candidates linked to peaks on Chr 4 were involved in seed development and maturation, floral meristem identity, photoperiod, and vernalization in soybeans, *Arabidopsis*, and other plants. *Glyma.04g124200*, candidate for peak-1, encodes an ABA-responsive element binding protein with a bZIP domain (AREB3) that regulates diverse developmental processes during seed development and maturation in soybean, grapes, and other legumes [95–97]. *Glyma.04g125700*, also a candidate of peak-1, encodes a member of the Myb family of transcription factors (MYB33). MYB transcription factors play an important role in flowering time in *Arabidopsis* by directly repressing a flowering locus (*FT*) expression in the leaf [98]. The ortholog gene in tomato, *SIMYB33*, also regulated flowering and maturity by modulating the expression of genes responsible for flowering and sugar metabolism [99]. The other candidate for peak-2, *Glyma.04g163100*, is a PHD finger protein that regulates expression of floral repressors

related to photoperiod and vernalization pathways in *Arabidopsis*. It has been shown that mutations in the *PHD* genes cause increased expression of flowering genes and early flowering [100, 101].

The peak-3 on Chr 10 associated with R1, is located in a region that overlaps with previously reported QTLs for R1, R8, RL, seed protein, seed weight, and other disease resistance QTLs according to the SoyBase.org browser and previous studies [79, 80, 102]. We identified two potential candidate genes, *Glyma.10g180600* and *Glyma.10g182300*, located 33 and 105 kb apart from the significant SNP on Chr 10. *Glyma.10g180600* encodes a *Cryptochrome 2* (*CRY2a*) gene, a blue light receptor whose orthologs in tomato and *Arabidopsis* regulate leaf senescence, vegetative development, and R1. In soybeans, there are at least six cryptochrome genes in the soybean genome some of which influence flowering time. Previous research found that the *CRY2a* gene regulates leaf senescence in soybean by repressing a transcription activator (CIB1) that activates the transcription of senescence-associated genes [103–105]. Our other candidate gene, *Glyma.10g182300*, encodes a constitutively photomorphogenic 1 (COP1) interacting protein-related, whose ortholog in *Arabidopsis* is a central regulator of photoperiodic flowering [106, 107]. In this study, the association peak-3 showed to be highly influenced by the environment and further studies are needed to verify its role in flowering time.

In association studies, candidate genes or mechanisms underlying a trait are not directly revealed [108]; therefore, post-GWAS analysis is critical to determining relevant conclusions. A common strategy in the post-GWAS analysis is to consider the linkage disequilibrium around tagging SNPs to delimit a region of exploration for potential causal genes; however, those regions are usually large, and depending on their size, we can find hundreds or thousands of genes, which could make the identification of causal genes less accurate and more complex. To shorten the list of potential candidates, we used a methodology known as the "Synthetic phenotype association study" (SPAS), which provides a tool to measure the relationship between tagging variants from GWAS and genomic variants within nearby genes [73]. Overall, we found two stable peaks significantly associated with R8 and RL in a soybean panel sharing the same haplotype *e1-asE2E3*, where *GmFulb* (H38L) is the most significant candidate gene; however, further investigation of the specific candidate gene responsible for the phenotypic variation, as well as their underlying functional mechanisms remains to be done.

## Conclusions

This study demonstrates that characterizing populations sharing the same genotype for known *E* genes can help to identify additional loci affecting R1, R8, and RL. We identified two stable peaks that can promote or delay by two days R8 and RL, in addition to one significant SNP for R1 which was highly influenced by the environment. The detected peaks did not exhibit pleiotropic control over R1 and R8, suggesting that trait-specific minor effect loci are also involved in controlling R1 and R8, and that major *E* genes may regulate them. The independent loci acting over R1 and R8 also indicate that we could select soybean with earlier maturity independent of R1. From the total of 111 *Glyma* genes that showed high association with the three significant peaks in this study, we identified six genes that may play important roles in regulating R1, R8, and RL. Of them, *GmFulb* is the most compelling candidate given its previously reported functions. For future validation studies, parents differing for the peaks, with the most contrasting R8 phenotype, can be selected from the diversity panel to develop biparental populations. QTL flanking markers represent a valuable tool for soybean molecular breeding; however, fine mapping and map-based cloning studies of the candidate genes are necessary before

they can be used effectively to breed cultivars with optimal phenological traits. In addition, further studies comparing individuals with different genotypes of major *E* genes could increase our understanding of the role of minor-effect loci in soybean flowering and maturity.

## Supporting information

**S1 Fig. Map showing the four locations where flowering (R1), maturity (R8), and reproductive length (RL) measurements were taken from 2017 to 2020.** Orange squares delimit MG III and IV adaptation regions.
(TIF)

**S2 Fig. Heatmap from a genomic relationship matrix of the three hundred twenty-nine *G. max* USDA accession used in this study.**
(TIF)

**S3 Fig. Average squared correlation of allele frequencies ($r^2$) against distance across the whole genome from heterochromatic (red) and euchromatic regions (blue).**
(TIF)

**S4 Fig. Histograms of the frequency distribution of flowering (R1), maturity (R8), and reproductive length (RL) for three hundred twenty-nine *G. max* accessions.** Data are measurements of R1 and R8 in days after planting (DAP), and RL as the number of days between R1 and R8 from four years and four locations.
(TIF)

**S5 Fig. Optimal number of clusters by k-means using the average silhouette width.**
(TIF)

**S6 Fig. SNP density plot across 20 chromosomes of soybean representing the number of SNPs within 1 Mb window size.**
(TIF)

**S7 Fig. Significant SNPs from genome-wide association analysis by individual environments, years, and across all environments.** R1 is flowering time, R8 is maturity time, and RL is reproductive length. The plot is color coded by the effect of the SNPs for each trait. SNP positions are based on the Wm82.a2.v1 genome assembly.
(TIF)

**S8 Fig. Phenotypic differences between lines carrying the reference allele (Ref_Allele) and the alternative allele (Alt_Allele) of the seven tagging SNPs associated with R1, R8, and RL.** Bar plots show the differences in R8 (A), RL (B), and R1 (C). **** indicates significant differences at a *p*-value $\leq$ 0.0001 between the two groups. DAP is days after planting.
(TIF)

**S1 Table. Planting dates of experimental environments.**
(PDF)

**S2 Table. Mean flowering (R1), and maturity (R8) time in days after planting (DAP) and mean reproductive length (RL) in number of days per accession across all environments.**
(PDF)

**S3 Table. Descriptive statistics of phenotypic variation, genotypic variance (G) and broad-sense heritability ($H^2$) of days to flowering (R1), days to maturity (R8), and reproductive length for 329 *G. max* USDA accessions evaluated at nine environments.**
(PDF)

**S4 Table. Percentage of SNPs by heterochromatic and euchromatic regions of each chromosome.**
(PDF)

**S5 Table. Linkage Disequilibrium (LD) decay rate across 20 chromosomes within euchromatic and heterochromatic regions.**
(PDF)

**S6 Table. Summary of the single-nucleotide polymorphisms (SNPs) significantly associated with flowering time (R1), maturity time (R8), and reproductive length (RL) in three hundred twenty-nine G. max accessions across environments and years.**
(PDF)

**S7 Table. Summary of the significant single-nucleotide polymorphisms (SNPs) associated with flowering time (R1), maturity time (R8), and reproductive length (RL) in three hundred twenty-nine G. max accessions by individual environments, years, and across all environments and years.**
(PDF)

**S8 Table. List of potential candidate genes by tagging SNPs with average accuracy and descriptions for flowering time (R1), maturity time (R8), and reproductive length (RL).**
(PDF)

**S9 Table. Correspondence between tagging SNPs and known E genes measured by average accuracy obtained from AcuTool.**
(PDF)

**S1 File.**
(DOCX)

## Acknowledgments

We express our gratitude to GDM Seeds, Inc. for providing an experimental location.

## Author Contributions

**Conceptualization:** Diana M. Escamilla, Nicholas Dietz.

**Data curation:** Diana M. Escamilla.

**Formal analysis:** Diana M. Escamilla.

**Funding acquisition:** Katy Martin Rainey.

**Investigation:** Diana M. Escamilla, Nicholas Dietz.

**Methodology:** Katy Martin Rainey.

**Project administration:** Katy Martin Rainey.

**Resources:** Kristin Bilyeu, Katy Martin Rainey.

**Supervision:** Kristin Bilyeu, Katy Martin Rainey.

**Writing – original draft:** Diana M. Escamilla.

**Writing – review & editing:** Kristin Bilyeu, Karen Hudson, Katy Martin Rainey.

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
