## [Decision Letter · Decision Letter 0]

11 May 2023

PONE-D-23-10490Genome-wide association study revealed GmFULb as a candidate gene controlling maturity and reproductive lengthPLOS ONE

Dear Dr. Rainey,

Thank you for submitting your manuscript to PLOS ONE. After careful consideration, we feel that it has merit but does not fully meet PLOS ONE’s publication criteria as it currently stands. Therefore, we invite you to submit a revised version of the manuscript that addresses the points raised during the review process.

We look forward to receiving your revised manuscript.

Kind regards,

Gunvant Patil, PhD

Academic Editor

PLOS ONE

Journal Requirements:

Additional Editor Comments (if provided):

Thank you for submitting your manuscript to PLOS One. The editorial team and a group of expert reviewers have assessed your submission and feel that it has potential for publication, and so we would like to invite you to revise the paper and resubmit for further review.

We appreciate that your paper addresses each and every comments/questions. Please see the attached reviewer comments for further details about necessary revisions. Please note, your revised manuscript should be accompanied by a summary of your responses to the reviewers' comments. You have 4 weeks to respond to this revise and resubmit request ending on June 15, 2023 after which point we will presume that you have withdrawn your submission.

Please feel free to contact me with any questions.

Reviewers' comments:

Reviewer's Responses to Questions

**Comments to the Author**

1. Is the manuscript technically sound, and do the data support the conclusions?

Reviewer #1: Yes

Reviewer #2: Yes

2. Has the statistical analysis been performed appropriately and rigorously? 

Reviewer #1: Yes

Reviewer #2: Yes

3. Have the authors made all data underlying the findings in their manuscript fully available?

Reviewer #1: Yes

Reviewer #2: Yes

4. Is the manuscript presented in an intelligible fashion and written in standard English?

Reviewer #1: No

Reviewer #2: No

5. Review Comments to the Author

Reviewer #1: The authors have performed GWAS analyses in soybean and documented some interesting finding. However, the paper is poorly drafted, and the findings are not well presented.

I have few suggestions,

1) Make title precise. Include the word "soybean" in the title.

2) Please narrow down the introduction section to your focused research question or hypothesis.

3) Line 161. Which package in R? Add a reference if applicable.

4) Avoid short forms in the abstract. Chr=Chromosome.

5) Throughout the manuscript, editing of English language required.

6) Line 245-246. How were the genes selected to be further analyzed? The statements are unclear.

7) Make discussion section short and precise. The explanation of results should go to the result section.

8) Line 281-282. Are you clear on the statement? Even 5,000 markers are sufficient. Please verify the statement and reference.

9) I suggest redrafting the GWAS section. It's a little confusing

Reviewer #2: Comments:

1. Line 130-133: Are there any differences in R1 for the maturity checks across locations? The photoperiod highly influences R1 in soybean. As the experimental sites varied between the years and the planting times also varied, environmental variation and germination should affect the R1. In that case, mapping R1 by combining all the years of data may not be appropriate. The authors also analyze the R1 data on an environmental basis-which should be the right way. In GWAS studies, there are some variations in germination time after planting depending on genotype, soil type, and weather in that particular year.

2. Line 197-202: This is not a criticism, but rather a query. Why the authors calculated the narrow sense heritability? From breeders' perspective, narrow sense heritability is vital to estimate the phenotypic variance attributed to the additive effects in the progeny developed from two parents. The GWAS studies usually measure the mean entry basis broad sense heritability. The authors used the kinship coefficients (K) to calculate the narrow sense heritability, which makes sense. Still, my query is, what are the implications of the narrow sense heritability in a GWAS study? What additional information can it provide us about the trait or the environment where the traits were tested?

3. Line 262-264: The observed broad sense heritability is almost identical (R1) or lower (R8 and RL) compared to the narrow sense heritability. Usually, the trend is the opposite- observed narrow sense heritability should be lower than the broad sense heritability because narrow sense heritability is related to the phenotypic variance only caused by additive effect (a portion of genetic variance). In addition, I am asking what additional information the h2 is providing us is. The authors did not discuss the heritability much in the discussion section.

4. Line 265-266: I'm afraid I have to disagree. In Figure 1, the dot (uppermost left side) represents the genotype, and the dot (right lowermost side) "MAY be" part of the different hidden subpopulations. I would suggest performing a cluster analysis (for example, using mclust in R) and seeing how many clusters there are using the first two principal components. That can provide information if there is a hidden stratification within the population.

5. Line 287: The authors refer to two different QTL as "Two strong peaks", right?

6. Line 316: The candidate gene exploration technique is very comprehensive. It's a good demonstration of using the tools in SoyKb. I would be more interested to see how these tools perform to narrow down the candidate genes for quantitative abiotic stress-tolerant traits.

7. Line 437-438: As my first comment-highly influenced by the environment. It would be interesting to see the checks are varying between the environments.

8. Line 461: Again-two peaks on Gm04 are two different QTL , right? Because they are further apart in physical distance.

9. Line 537: Please insert references for the sentence "The previously reported function of Glyma.10g182300…..and RL”

10. Line 564: GmFulb is the candidate for Peak-2 on Gm04, not a candidate for both peaks. The two peaks are further apart in their physical position and should be regarded as two different loci.

6. PLOS authors have the option to publish the peer review history of their article (what does this mean?). If published, this will include your full peer review and any attached files.

Reviewer #1: No

Reviewer #2: No

---

## [Author Response · Author response to Decision Letter 0]

20 Sep 2023

**Note that the text below is copied from the Response to Reviewers document:

Dear Editor and Reviewers,

Thank you so much for all the comments made to the manuscript. They have been very helpful to improve the quality of the manuscript and ensure readers can easily follow what we are presenting in this study. Please find below all the comments received from you and the response from us. We also uploaded a new revised version of the manuscript, a new supplementary figure, and an updated supplementary material file. Everything is with tracked changes for you to see what was modified.

Reviewer #1: The authors have performed GWAS analyses in soybean and documented some interesting findings. However, the paper is poorly drafted, and the findings are not well presented.

I have few suggestions,

1) Make title precise. Include the word "soybean" in the title.

Response: Title was modified to make it more precise, and the word soybean was included.

2) Please narrow down the introduction section to your focused research question or hypothesis.

Response: The introduction was narrowed down as suggested. 

3) Line 161. Which package is R? Add a reference if applicable.

Response: We didn’t use an R package. For the PCA we used the built-in function, eigen, in the R software. With this function we did the eigen decomposition of the marker matrix and extracted the PCs. 

4) Avoid short forms in the abstract. Chr=Chromosome.

Response: Chr was replaced by chromosome in the abstract. 

5) Throughout the manuscript, editing of English language required.

Response: The manuscript was revised and edited as required. 

6) Line 245-246. How were the genes selected to be further analyzed? The statements are unclear.

Response: This section was revised and rewrite to improve the clarity of the statements. 

7) Make the discussion section short and precise. The explanation of results should go to the result section.

Response: The discussion section was edited as suggested and explanation of results was removed from the discussion. 

8) Line 281-282. Are you clear on the statement? Even 5,000 markers are sufficient. Please verify the statement and reference.

Response: Thanks for pointing this out. After reading the statement again we feel it could be misleading. It was rewritten to bring more clarity, emphasizing that with as few as 9,600 markers we could capture the haplotype variation present in euchromatic regions of the soybean genome, and that this of course can vary depending on the LD present in the studied populations. It’s also important to note that this statement and reference is specific to soybean populations.

9) I suggest redrafting the GWAS section. It's a little confusing.

Response: the results GWAs section was redraft to provide more clarity to the readers. 

Reviewer #2: Comments:

1. Line 130-133: Are there any differences in R1 for the maturity checks across locations? The photoperiod highly influences R1 in soybean. As the experimental sites varied between the years and the planting times also varied, environmental variation and germination should affect the R1. In that case, mapping R1 by combining all the years of data may not be appropriate. The authors also analyze the R1 data on an environmental basis-which should be the right way. In GWAS studies, there are some variations in germination time after planting depending on genotype, soil type, and weather in that particular year.

Response: 

As observed below, there were significant differences in R1 across locations for the checks. Yes, we agree that R1 is highly influenced by the environment and that analyzing R1 data on an environmental basis is important. However, when running the GWAs across environments, we first obtained the BLUPs for each trait using the mixed linear model approach where we accounted for the environmental effect (location, year), which allow us to also perform the GWAs across all environments. The environmental (location x year) effect will encompass all these differences in soil type, planting time, and the specific weather conditions. 

summary(aov(fit_R1))

 Df Sum Sq Mean Sq F value Pr(>F) 

Accession 4 3481 870 88.89 <2e-16 ***

Year:Location 6 27561 4593 469.13 <2e-16 ***

Residuals 650 6364 10 

---

Signif. codes: 0 ‘***’ 0.001 ‘**’ 0.01 ‘*’ 0.05 ‘.’ 0.1 ‘ ’ 1

summary(aov(fit_R8))

 Df Sum Sq Mean Sq F value Pr(>F) 

Accession 4 6917 1729 64.98 <2e-16 ***

Year:Location 6 57260 9543 358.60 <2e-16 ***

Residuals 649 17272 27 

---

Signif. codes: 0 ‘***’ 0.001 ‘**’ 0.01 ‘*’ 0.05 ‘.’ 0.1 ‘ ’ 1

> summary(aov(fit_RL))

 Df Sum Sq Mean Sq F value Pr(>F) 

Accession 4 6703 1675.9 80.28 <2e-16 ***

Year:Location 6 8383 1397.2 66.93 <2e-16 ***

Residuals 649 13548 20.9 

---

Signif. codes: 0 ‘***’ 0.001 ‘**’ 0.01 ‘*’ 0.05 ‘.’ 0.1 ‘ ’ 1

2. Line 197-202: This is not a criticism, but rather a query. Why the authors calculated the narrow sense heritability? From breeders' perspective, narrow sense heritability is vital to estimate the phenotypic variance attributed to the additive effects in the progeny developed from two parents. The GWAS studies usually measure the mean entry basis broad sense heritability. The authors used the kinship coefficients (K) to calculate the narrow sense heritability, which makes sense. Still, my query is, what are the implications of the narrow sense heritability in a GWAS study? What additional information can it provide us about the trait or the environment where the traits were tested?

Response: Thanks for pointing this out. You are right for GWAs studies, mean entry basis broad sense heritability is more commonly used because it gives us an idea of how much of the trait’s performance is due to genetics and what is due to other environmental factors. We also estimated the broad sense heritability. In reality, the narrow sense heritability was just extra information, so to avoid confusion and not presenting irrelevant data in the discussion section, we removed the narrow sense heritability from the main text and just leave that as extra information in the supplementary material. 

3. Line 262-264: The observed broad sense heritability is almost identical (R1) or lower (R8 and RL) compared to the narrow sense heritability. Usually, the trend is the opposite- observed narrow sense heritability should be lower than the broad sense heritability because narrow sense heritability is related to the phenotypic variance only caused by additive effect (a portion of genetic variance). In addition, I am asking what additional information the h2 is providing us is. The authors did not discuss heritability much in the discussion section.

Response: Thanks for pointing this out. Initially, to estimate the broad sense heritability, we used the model from equation 1 where we fit the environment (location x year), block and spatial covariate effects. To calculate the broad sense heritability, we used the variance due to the genotypes and the residual variance of the model from equation 1. When we went to estimate the narrow sense heritability the model only used the BLUPs and marker data, where we get an additive and residual variance. However, this residual variance would likely only include non-additive genetic variance; thus, being different from the residual variance used in equation 1. for the broad sense heritability, the residual variance used, will likely include the variance due to GxE. The residual variance was adjusted (Ve first model + Ve second model) to estimate the narrow sense heritability and values were corrected in supplementary Table S3. Another alternative to do this is to run a single step model, instead of two steps. 

Heritability is a population parameter, that defines how much of the phenotypic variation is due to genetic values and what is due to other factors such as the environment. So, the purpose of reporting it in the paper is more as a reference parameter of the studied populations where readers can have a sense of how much is the environment affecting the studied traits. And also, how much of this trait performance is controlled by genetics, considering this population is fixed for three major E genes. We also edited the text to include information about broad sense heritability when appropriate. 

4. Line 265-266: I'm afraid I have to disagree. In Figure 1, the dot (uppermost left side) represents the genotype, and the dot (right lowermost side) "MAY be" part of the different hidden subpopulations. I would suggest performing a cluster analysis (for example, using mclust in R) and seeing how many clusters there are using the first two principal components. That can provide information if there is a hidden stratification within the population.

Response: We performed cluster analysis and found that the optimal number of clusters was two. However, we want to acknowledge that to conduct the GWAS, we used an empirical Bayesian framework using the R package NAM, whose model includes a polygenic term that accounts for the population structure to minimize false positives and increase statistical power. Description of the population structure of the population was corrected adding the extra information generated by the clustering analysis. The figure of Optimal number of clusters by k-means using the average silhouette width was added as supplementary figure S2. Methodology was also updated to include the clustering analysis. 

5. Line 287: The authors refer to two different QTL as "Two strong peaks", right?

Response: Yes, the GWAs peaks can be considered two different QTLs given the distance between them. To maintain consistency across the document we referred to them as Peak-1, Peak-2 and Peak-3. 

6. Line 316: The candidate gene exploration technique is very comprehensive. It's a good demonstration of using the tools in SoyKb. I would be more interested to see how these tools perform to narrow down the candidate genes for quantitative abiotic stress-tolerant traits.

Response: Yes, it was a very interesting approach that helped us to do a more systematic exploration of the significant peaks (QTLs) detected in this study. It’s very user friendly and by reading Škrabišová et al. 2022, you can find a more detailed description of this tool. 

7. Line 437-438: As my first comment-highly influenced by the environment. It would be interesting to see the checks vary between the environments.

Response: Information can be found in the previous question. 

8. Line 461: Again-two peaks on Gm04 are two different QTL , right? Because they are further apart in physical distance.

Response: Yes, it would be two different QTL, for keeping consistency we referred to them through the text as peaks. 

9. Line 537: Please insert references for the sentence "The previously reported function of Glyma.10g182300…..and RL”

Response: Reference was added

10. Line 564: GmFulb is the candidate for Peak-2 on Gm04, not a candidate for both peaks. The two peaks are further apart in their physical position and should be regarded as two different loci.

Thanks for pointing out that, we rewrote some sentences of those sections to specify the specific candidate genes for each peak (QTL).

---

## [Decision Letter · Decision Letter 1]

26 Oct 2023

Genome-wide association study reveals GmFULb as candidate gene for maturity time and reproductive length in Soybeans (Glycine max)

PONE-D-23-10490R1

Dear Dr. Rainey,

We’re pleased to inform you that your manuscript has been judged scientifically suitable for publication and will be formally accepted for publication once it meets all outstanding technical requirements.

Kind regards,

Gunvant Patil, PhD

Academic Editor

PLOS ONE

Additional Editor Comments (optional):

Reviewers' comments:

Reviewer's Responses to Questions

**Comments to the Author**

1. If the authors have adequately addressed your comments raised in a previous round of review and you feel that this manuscript is now acceptable for publication, you may indicate that here to bypass the “Comments to the Author” section, enter your conflict of interest statement in the “Confidential to Editor” section, and submit your "Accept" recommendation.

Reviewer #1: All comments have been addressed

2. Is the manuscript technically sound, and do the data support the conclusions?

Reviewer #1: Yes

3. Has the statistical analysis been performed appropriately and rigorously? 

Reviewer #1: Yes

4. Have the authors made all data underlying the findings in their manuscript fully available?

Reviewer #1: Yes

5. Is the manuscript presented in an intelligible fashion and written in standard English?

Reviewer #1: Yes

6. Review Comments to the Author

Reviewer #1: (No Response)

7. PLOS authors have the option to publish the peer review history of their article (what does this mean?). If published, this will include your full peer review and any attached files.

Reviewer #1: No

---

## [Editor Report · Acceptance letter]

10 Jan 2024

PONE-D-23-10490R1 

PLOS ONE

Dear Dr. Rainey, 

I'm pleased to inform you that your manuscript has been deemed suitable for publication in PLOS ONE. Congratulations! Your manuscript is now being handed over to our production team.

Kind regards, 

on behalf of

Dr. Gunvant Patil 

Academic Editor

PLOS ONE